# Learning to Think in Blocks: A Prior-Guided Reinforcement Learning Framework for RAG

## Abstract

Retrieval-Augmented Generation (RAG) systems mitigate factual inaccuracies in large language models (LLMs) by integrating external knowledge, but their effectiveness often hinges on query rewriting techniques. Prompt-based rewriting methods are frequently suboptimal, while existing reinforcement learning (RL) approaches struggle with inefficient, unguided exploration of the vast strategy space. To address these limitations, we propose an end-to-end RL framework that initializes the training process with human-defined prior rewriting strategies, enabling the model to learn from its interactions with the RAG environment and develop its own effective posterior rewriting strategies. Furthermore, we develop a novel RL algorithm, namely Block-wise Geometric Policy Optimization (BGPO), which resolves the granularity mismatch in previous methods by redefining the state-action space as *blocks* of tokens. This algorithm is enhanced by geometric averaging for balanced importance and a Bellman-equation-inspired credit assignment mechanism to reshape the reward. Extensive experiments on both local corpus retrieval and online search datasets demonstrate that our RL framework consistently surpasses the baselines, validating its superiority for complex RAG tasks. Our project code can be found at this anonymous repository.

## 1 Introduction

Retrieval-Augmented Generation (RAG) (Lewis et al., 2020) is a well-established paradigm enabling large language models (LLMs) to generate augmented content by retrieved external knowledge. However, for some complex queries, simple retrieval using the original query usually cannot return the expected information due to query complexity over embedding capability. Therefore, some prompt-based methods (Trivedi et al., 2023; Li et al., 2025) leverage specific techniques to rewrite the query for more effective retrieval, and some others methods based on reinforcement learning (RL) (Jin et al., 2025; Song et al., 2025) train the model itself to learn how to rewrite.

Despite their effectiveness, these methods for RAG tasks exhibit inherent limitations. First, prompt-based methods are often labor-intensive for sophisticated prompt design, and might be suboptimal as the models without training only have weak rewriting capability. Second, although RL methods can optimize the model performance through training, most current frameworks require the model itself to find out successful rewriting strategies without any guidance during the rollout phase. This mechanism lowers the efficiency of the rollout process because the potential rewriting space is vast and complex, making it hard for the model to explore alone.

To address these shortcomings, we propose a novel framework in this paper that integrates human-defined prior strategies to provide an effective starting point for RL process. In our framework, the model practices human prior rewriting strategies to interact with the environment. Then, through the process of RL, the model successfully learns how to correctly rewrite queries in a real RAG environment. Therefore, the model can start from the human prior, through the training in the RAG environment, and finally learn posterior rewriting strategies of its own. The human prior makes the interaction with the environment more efficient, and the learning process makes the model better understand the rewriting heuristics, the RAG environment and dataset. The overview of our framework is displayed in Figure 1.

To train our RAG system by RL more effectively, we specifically design a new RL algorithm with a redefined state-action granularity called Block-wise Geometric Policy Optimization (BGPO). The

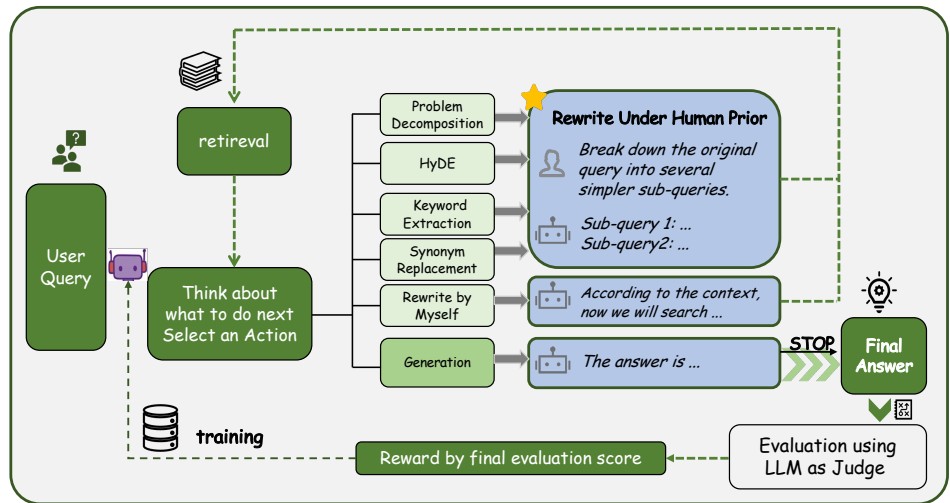

Figure 1: The overview of our Prior-Guided Reinforcement Learning framework for RAG.

state-action granularity definitions in most current RL algorithms are suboptimal for multi-turn and conversational-like reasoning tasks. These methods typically model actions at either the token level (e.g., the overly fine-grained GRPO (Shao et al., 2024), DAPO (Yu et al., 2025)), or the full sequence level (e.g., the too coarse GSPO (Zheng et al., 2025)). Token-level granularity creates a noisy, high-variance learning signal while sequence-level granularity applies a uniform learning signal that fails to distinguish between tasks requiring complex long trajectories and those needing simple short ones. To overcome this problem, we develop a new RL algorithm BGPO, which redefines the state and action spaces. Specifically, our BGPO algorithm defines each conversational turn as a "block" and the generation of such "block" as an action, which structures states and actions at a granularity more suitable than the traditional token or sequence level.

To promote our algorithm further, we optimize the reward shaping in RL algorithms. Most of current RL algorithms applied in RAG assign equal advantage on every token, which may not be beneficial for RAG settings. For RAG, first-round query understanding and query rewriting may be much more important than final generation which is often a simple summary of retrieved documents. Therefore, we extend the discounting factor of Bellman equation (Bellman, 1957) to an emphasis factor, which exponentially increases the advantage for the beginning stage of RAG reasoning process.

In summary, our contributions in this paper are as follows:

1. We propose an end-to-end RL framework for RAG tasks that enables a model to initialize its policy from human-provided prior knowledge, and subsequently learn a refined posterior distribution of rewriting strategies through environmental interaction.

2. In our framework, we develop Block-wise Geometric Policy Optimization (BGPO) algorithm, tailored for multi-step reasoning, which introduces a block-level state-action definition to create a more suitable granularity than traditional token- or sequence-level approaches.

3. Specifically, we further propose a novel reward-shaping technique tailored for RAG. By reversing the logic of traditional temporal discounting, we apply an emphasis factor to prioritize and reward the crucial, initial reasoning steps of the agent more heavily.

## 2 RELATED WORK

Retrieval-Augmented Generation (RAG) (Lewis et al., 2020; Guu et al., 2020; Hu & Lu, 2024) enhances language model outputs by incorporating external knowledge, improving factuality and timeliness. Existing RAG systems often rely on fixed, heuristic-based query rewriting techniques, such as HyDE (Gao et al., 2023) and question decomposition (Ammann et al., 2025), which cannot adapt to specific data distributions. To address this, we propose an end-to-end RL framework that

enables the model to autonomously learn query rewriting policies directly from interactions with the RAG environment.

Prior RL-based methods like Search-R1 (Jin et al., 2025) and R1-Searcher (Song et al., 2025) train RAG agents for iterative query refinement but place the full exploration burden on the base model, which is inefficient given the vast natural language action space. We address this by incorporating pre-defined rewriting strategies as a human-informed prior. The model initially leverages these heuristics and then refines its policy via RL on the target dataset, allowing it to combine effective priors with novel, data-driven strategies, improving both learning efficiency and overall performance.

Policy optimization for language models has progressed from Proximal Policy Optimization (PPO) (Schulman et al., 2017) to methods like GRPO (Shao et al., 2024), DAPO (Yu et al., 2025), and GSPO (Zheng et al., 2025), which successively simplified architectures and improved stability. Building on this, we propose a novel algorithm for multi-turn RAG that bridges GSPO's coarse, sequence-level credit assignment and PPO's fine-grained, token-level approach. By defining "semantic blocks" as action units, computing importance sampling ratios geometrically, and applying a Bellman-inspired discount to advantage estimates, our hierarchical credit assignment enhances stability and efficiency in complex, multi-step reasoning tasks.

## 3 METHODOLOGY

### 3.1 REINFORCEMENT LEARNING FRAMEWORK FOR RAG

In brief, our proposed framework integrates human expertise into a query rewriting process through modeling an RAG task as a sequential decision-making problem by RL. In the framework, we construct an agent with an LLM, which learns to navigate a hybrid action space to iteratively refine its search strategy based on retrieved information.

For a given user query $q_{\text{user}}$, the RL process begins with an initial state $s_0$, which contains the system prompt and $q_{\text{user}}$. At each turn $t$, the agent operates within the state $s_t$ that represents the full conversational history. Inspired by the ReAct paradigm Yao et al. (2023), the agent follows an iterative cycle consisting of three steps: reasoning, acting, and observing.

**Hybrid Action Space** The core of our framework is the hybrid action space $\mathcal{A}$, which is composed of three distinct categories of actions. The first category, **Human-Prior Heuristics** ($\mathcal{A}_H$), comprises a curated set of four pre-defined rewriting strategies: *decomposition*, *keyword_extraction*, *synonym_replacement*, and *HyDE* (Gao et al., 2023). These strategies are introduced to the model via concise non-prescriptive descriptions in the system prompt, to encourage learning through interaction. The second category, **Self-Devised Rewriting** ($\mathcal{A}_M$), provides a single flexible action *rewrite_by_myself*, empowering the model to generate novel queries when the pre-defined heuristics are insufficient. The final category, **Terminal Action** ($\mathcal{A}_T$), contains the *generation* action to conclude the iterative process and synthesize the final answer. The complete action space is thus defined as the union of these categories: $\mathcal{A} = \mathcal{A}_H \cup \mathcal{A}_M \cup \mathcal{A}_T$.

**Multi-Turn Reasoning and Retrieval Cycle** The agent's interaction with the environment is structured as a multi-turn conversation. At each turn $t$, the cycle begins with a **reasoning** step given the current state $s_t$, where the agent generates an internal monologue to analyze the gathered information and determines the best course of action. The reasoning step yields the selection of an action $a_t \in \mathcal{A}$ based on the learned policy $\pi(a_t|s_t)$. Following the selection, the chosen action $a_t$ is **executed**. If it is a rewriting strategy ($a_t \in \mathcal{A}_H \cup \mathcal{A}_M$), the model generates a new (rewritten) query $q_t$, leading to the final step of the cycle: **observation**. To provide immediate feedback, the rewritten query $q_t$ is used to retrieve a set of documents $D_t = \{d_1, d_2, \ldots, d_k\}$ from the external knowledge corpus $\mathcal{C}$. Then, these retrieved documents are appended to the conversational history, forming the new state $s_{t+1} = s_t \oplus (\text{retrieved\_docs: } D_t)$, where $\oplus$ denotes concatenation. This feedback loop allows the agent to dynamically adjust its subsequent actions. The entire retrieval cycle repeats until the agent selects the terminal action $a_t = \textit{generation}$, where it synthesizes a final answer based on the complete history of queries and retrieved documents accumulated throughout the episode.

### 3.2 REINFORCEMENT LEARNING ALGORITHM OF BGPO

In this subsection, we detail our proposed BGPO algorithm. Firstly, we redefine the state and action space in RL. In a trajectory consisting of a sequence of tokens $y = (y_1, y_2, \ldots, y_T)$, most of previous methods (Schulman et al., 2017; Shao et al., 2024; Yu et al., 2025) define the state at timestep $t$ as the preceding token sequence $s_t = y_{<t}$, and the action $a_t$ as the next chosen token $y_t$ from the vocabulary. However, in our multi-turn RAG reasoning setting, the semantic impact of a single token is often minimal. Instead, a coherent "block" of thought, such as a rewritten query or a step in a reasoning chain, is indeed the foundation that influences the trajectory's outcome.

Formally, we define a "block" as a single, complete turn of the Assistant's response within the multi-turn dialogue trajectory. Unlike token-level RL, we treat the entire Assistant turn as an atomic unit. A block explicitly comprises: (1) Internal Monologue (Chain-of-Thought reasoning), (2) Action Decision (the selected strategy), and (3) Execution Content (the rewritten query or answer). We apply masking to System and User nodes, targeting only these unmasked Assistant blocks for policy optimization.

Under this definition, we partition the trajectory $y$ into $K$ semantic blocks, $y = (b_1, b_2, \ldots, b_K)$. We define the state at step $k$ as the sequence of preceding blocks, denoted as $s_k = (b_1, \ldots, b_{k-1})$, and the action as the generation of the entire next block, denoted as $a_k = b_k$. Then, the policy for generating a block is the product of the probabilities of its tokens, formulated as $\pi_\theta(b_k|s_k) = \prod_{t=1}^{|b_k|} \pi_\theta(y_{k,t}|s_k, y_{k,<t})$, where $y_{k,t}$ is the $t$-th token in block $b_k$.

Recall that the PPO's optimizing objective (Schulman et al., 2017) is defined at token level, using a clipped importance ratio and a fine-grained advantage estimate $\hat{A}_t$ from Generalized Advantage Estimation (GAE) as:

$$\mathcal{J}_{\text{PPO}}(\theta) = \hat{\mathbb{E}}_t \left[ \min \left( w_t(\theta)\hat{A}_t, \text{clip}(w_t(\theta), 1 - \epsilon, 1 + \epsilon)\hat{A}_t \right) \right],$$

where $w_t(\theta) = \frac{\pi_\theta(y_t|s_t)}{\pi_{\theta_{old}}(y_t|s_t)}$. In order to save computational resources, GRPO (Shao et al., 2024) replaces the value model with a group sampling estimation as follows, where the advantage is calculated at the sample level and distributed uniformly across all tokens (Shao et al., 2024):

$$\mathcal{J}_{\text{GRPO}}(\theta) = \mathbb{E}_{\substack{x \sim \mathcal{D}, \\ \{y_i\}_{i=1}^G \sim \pi_{\theta_{old}}}} \left[ \frac{1}{G} \sum_{i=1}^G \frac{1}{|y_i|} \sum_{t=1}^{|y_i|} \min \left( w_{i,t}(\theta)\hat{A}_i, \text{clip}(w_{i,t}(\theta), 1 - \epsilon, 1 + \epsilon)\hat{A}_i \right) \right].$$

Here, the sample advantage $\hat{A}_i = \frac{r(x, y_i) - \text{mean}(\{r(x, y_j)\})}{\text{std}(\{r(x, y_j)\})}$ is uniquely attributed to all tokens in sample $y_i$ equivalently.

At first, we investigate the difference between the two methods: PPO's credit assignment is quite fine-grained (per-token), which is computationally burdensome due to the need for a value model. Conversely, GRPO's credit assignment is very coarse-grained (per-sample), diminishing the varying importance of different reasoning steps. For instance, initial query understanding and rewriting steps are often more critical than later summarization steps. Therefore, inspired by the Bellman equation (Bellman, 1957), we propose a novel **reward-shaping** technique. To be specific, we introduce an emphasis factor $\gamma \in (0, 1]$ to attribute credit at block level. The advantage for any token within block $b_k$ of sample $y_i$ is reshaped as:

$$\hat{A}_{i,k} = \gamma^{k-1}\hat{A}_i.$$

The factor effectively shapes the reward landscape, prioritizing foundational reasoning steps with exponentially greater weight. It provides a more nuanced credit assignment than GRPO without the computational overhead of PPO's value model.

Secondly, we refer to GSPO (Zheng et al., 2025) which improves GRPO by correcting the importance sampling ratio to operate at sequence level, thereby stabilizing training (Zheng et al., 2025). The GSPO's objective uses a sequence-level ratio $s_i(\theta)$ as:

$$\mathcal{J}_{\text{GSPO}}(\theta) = \mathbb{E}_{\substack{x \sim \mathcal{D}, \\ \{y_i\}_{i=1}^G \sim \pi_{\theta_{old}}}} \left[ \frac{1}{G} \sum_{i=1}^G \min \left( s_i(\theta)\hat{A}_i, \text{clip}(s_i(\theta), 1 - \epsilon, 1 + \epsilon)\hat{A}_i \right) \right],$$

where $s_i(\theta) = \left(\frac{\pi_\theta(y_i|x)}{\pi_{\theta_{old}}(y_i|x)}\right)^{1/|y_i|}$. While GSPO's sequence-level importance ratio is a theoretical improvement, its sample-level objective calculation assigns equal weight to each sample, regardless of length. However, as introduced in DAPO (Yu et al., 2025), this ratio can down-weight the contribution of tokens in longer and more complex reasoning chains, as this approach fails to distinguish between complex tasks requiring long-form answers, which are crucial in RAG settings, and simple tasks where such length constitutes an undesirable pattern of verbosity. A token-level policy gradient loss, which is normalized by the total number of tokens in a batch, is more suitable as it ensures each token contributes equally to the gradient update.

Therefore, we adopt a token-level loss structure in our algorithm. Furthermore, we refine the importance ratio calculation for our block-wise setting. As a geometric mean over all tokens, the original GSPO ratio implicitly weights blocks by their token length. We argue that each reasoning block should contribute equally to measure how "off-policy" a sequence is. Thus, we propose a hierarchical importance ratio that first computes a geometric mean of token ratios **within** each block, and then computes a geometric mean of these block-level ratios. Formally, for a sample $y_i$ composed of $K_i$ blocks, the ratio of block $b_k$ and the final hierarchical ratio for the entire sequence are:

$$s_{i,k}(\theta) = \left(\frac{\pi_\theta(b_k|s_k)}{\pi_{\theta_{old}}(b_k|s_k)}\right)^{\frac{1}{|b_k|}}, \quad s'_i(\theta) = \left(\prod_{k=1}^{K_i} s_{i,k}(\theta)\right)^{\frac{1}{K_i}}.$$

This computation ensures each block is weighted equally. Inspired by DAPO, our Block-wise Geometric Policy Optimization (BGPO) objective combines this hierarchical ratio with our block-discounted advantage $\hat{A}_{i,k}$ within a token-level loss framework as:

$$\mathcal{J}_{\text{BGPO}}(\theta) = \mathbb{E}_{\substack{x \sim \mathcal{D}, \\ \{y_i\}_{i=1}^G \sim \pi_{\theta_{old}}}} \left[\frac{1}{\sum_{j=1}^G |y_j|} \sum_{i=1}^G \sum_{k=1}^{K_i} \sum_{t=1}^{|b_k|} \min\left(s'_i(\theta)\hat{A}_{i,k}, \text{clip}(s'_i(\theta), 1-\epsilon, 1+\epsilon)\hat{A}_{i,k}\right)\right].$$

Since our proposed modifications, including block-wise advantage assignment and a hierarchical importance ratio, are orthogonal to the other key techniques in DAPO, such as Clip-Higher and Dynamic Sampling, our method can be directly integrated into the DAPO framework to potentially yield further improvements.

We display the pseudo process of our BGPO algorithm in Algorithm 1.

## 4 EXPERIMENTS

### 4.1 DATASETS AND EVALUATION METRICS

We choose HotpotQA (Yang et al., 2018), 2WikiMultihopQA (Ho et al., 2020), MuSiQue (Trivedi et al., 2022), NQ (Kwiatkowski et al., 2019), TriviaQA (Joshi et al., 2017), PopQA (Mallen et al., 2023), Bamboogle (Press et al., 2023) as our datasets. From these datasets, we choose a filtered group of training sets of HotpotQA, 2WikiMultihopQA and MuSiQue as our in-domain dataset and the dev sets of all datasets as out-of-domain (OOD) datasets.

For the training set, we filter out noisy data (instances with ambiguous logic or missing evidence) to prevent reward hacking. However, we strictly maintain data complexity: 23.95% of our retained HotpotQA training data is labeled "hard", and 25.35% requires reasoning across 3+ supporting facts. Crucially, all evaluation results reported in Table 1 are based on the original, unfiltered test sets, ensuring our model is tested on the full range of difficulty. The detailed filtering process is introduced in Appendix A.

After training on the combined training dataset, we test the performance of the model on OOD datasets. For NQ and TriviaQA, we use the corpus provided from Karpukhin et al. (2020). For PopQA and Bamboogle, we directly use online search in Wikipedia instead of local corpus. Additionally, to address the complexity of real-world retrieval, we incorporate the SimpleQA benchmark and evaluate it using a real-time internet search engine API, moving beyond static corpora. The data statistics are listed in Table 4

---

**Algorithm 1** BGPO: Block-wise Geometric Policy Optimization

---

1: Initialize policy parameters $\theta$;
2: **for** training_step $= 1, 2, \ldots, M$ **do**
3:     $\theta_{old} \leftarrow \theta$;
4:     Initialize empty buffer $\mathcal{B}$;
5:     Sample a batch of queries $\{x_b\}_{b=1}^B$ from the dataset $\mathcal{D}$;
6:     **for** each query $x$ in the batch **do**
7:         Generate a group of $G$ responses $\{y_i\}_{i=1}^G$ using the frozen policy $\pi_{\theta_{old}}(\cdot|x)$;
8:         For each response $y_i$, parse it into blocks $(b_1, \ldots, b_{K_i})$;
9:         Compute rewards $r(x, y_i)$ and the sample-level advantage $\hat{A}_i$;
10:        Compute block-level advantages $\hat{A}_{i,k} = \gamma^{k-1}\hat{A}_i$ for $k = 1, \ldots, K_i$;
11:        Store the processed trajectories $(x, y_i, \{\hat{A}_{i,k}\}_{k=1}^{K_i})$ in buffer $\mathcal{B}$;
12:     **end for**
13:     **for** epoch $= 1, 2, \ldots, E$ **do**
14:         **for** each minibatch sampled from the buffer $\mathcal{B}$ **do**
15:             Compute hierarchical importance ratio $s_i'(\theta)$ for each sample in the minibatch;
16:             Compute the BGPO loss $\mathcal{J}_{\text{BGPO}}(\theta)$;
17:             Update policy parameters $\theta$ using gradient ascent: $\theta \leftarrow \theta + \alpha\nabla_\theta\mathcal{J}_{\text{BGPO}}(\theta)$;
18:         **end for**
19:     **end for**
20: **end for**

---

We use LLM-as-Judge as our evaluation metric. The base model for evaluation is Qwen3-32B (Yang et al., 2025) to get an accurate evaluation of the predictions. We take the probability of generating "Yes" as the first token within the output restriction of ["Yes", "No"] as the reward. The evaluation prompt for evaluation is contained in the Appendix B.

## 4.2 EXPERIMENT SETUP

We use the Low-Resource Reinforcement Learning (LSRL) (Liang, 2025) package to implement low-resource, high-throughput reinforcement learning using offloaded gradients. We use Qwen2.5-7B-Instruct (Yang et al., 2024) as the backbone of our framework, and run four A800 GPUs for approximately 2 days to complete the training process. For the retriever, we implement a hybrid retriever that combines sparse and dense retrieval. The sparse retrieval is based on BM25 (Robertson et al., 1994), and the dense retrieval is based on BGE-m3 (Chen et al., 2024). The weight between sparse and dense retrieval is set to 0.5, and consider top-3 retrieved documents. All baselines and our model use the same retriever and corpus.

**Training Details** For the training stage, the model is trained for a total of 3 epochs. For each data point, we sample 8 trajectories. The policy network is then trained on these collected samples using a learning rate of $1e - 6$ and a training batch size of 8. We apply gradient accumulation over 32 steps. The KL penalty coefficient $\beta$ is set to 0.001, and the upper clipping parameter is 0.28. During the generation phase for rollouts, we use a batch size of 32, a temperature of 0.9, and cap the tokens generated per round at 768, while the maximum model length is constrained to 5120 tokens. To optimize memory usage, we enable gradient offloading to the CPU. We use 1 GPU for the reward model, 1 GPU for vLLM rollouts, and 2 GPUs for data parallel (DP) training.

**Baselines** The Direct Answer method entails the model providing an immediate response to a question without showing its underlying reasoning. In contrast, the Chain-of-Thought (CoT) approach requires the model to first generate a step-by-step reasoning process before delivering the final answer. Naive RAG operates by having the model perform a single retrieval to gather relevant documents and subsequently generate a response based on that retrieved information. The IRCoT method (Trivedi et al., 2023) interleaves retrieval with the steps of a CoT, thereby using the reasoning to guide retrieval while the retrieved results concurrently improve the reasoning process. Finally, Search-o1 (Li et al., 2025) is a technique where the model analyzes the retrieved information for relevance and veracity before incorporating it into its reasoning chain. Search-R1 (Jin et al., 2025) is a

| Method | HotpotQA* | 2Wiki* | MuSiQue* | NQ† | TriviaQA† | Bamboogle‡ | PopQA‡ | Avg. |
|---|---|---|---|---|---|---|---|---|
| Direct Answer (7B) | 0.298 | 0.252 | 0.111 | 0.382 | 0.397 | 0.567 | 0.371 | 0.339 |
| CoT (7B) | 0.322 | 0.305 | 0.142 | 0.392 | 0.424 | 0.52 | 0.298 | 0.343 |
| NaiveRAG (7B) | 0.424 | 0.400 | 0.240 | 0.610 | 0.433 | 0.016 | 0.037 | 0.308 |
| IRCoT (7B) | 0.540 | 0.705 | 0.484 | 0.660 | 0.488 | 0.416 | 0.208 | 0.500 |
| Search-o1 (7B) | 0.526 | 0.694 | 0.396 | 0.554 | 0.673 | 0.440 | 0.263 | 0.506 |
| Search-R1 (14B) | 0.873 | **0.792** | **0.654** | 0.759 | **0.797** | **0.584** | **0.437** | **0.699** |
| RAG-BGPO (7B) | **0.880** | 0.780 | 0.611 | **0.800** | 0.755 | 0.568 | 0.430 | 0.689 |

Table 1: Performance comparison of different methods on various benchmarks. The best performance is set in bold and the second best performance is underlined. ∗/†/‡ means in-domain, out-domain, the mixture of out of domain and online search respectively.

RL framework combining reasoning and retrieval. Note that only Search-R1 is based on 14B model and other baselines are all based on 7B models.

## 4.3 RESULTS

The experimental results presented in Table 1 demonstrate the effectiveness of our proposed framework, denoted as **RAG-BGPO (7B)**. Due to the limitation of computational resources, we only train a 7B model, but it has comparable performance with Search-R1 based on 14B model. Compared with a range of baselines and state-of-the-art models, our approach shows strong performance across various QA benchmarks. Its advantages are particularly notable on complex multi-hop datasets requiring sophisticated reasoning. On HotpotQA, for instance, our model achieves 0.880, surpassing all 7B models and even the 14B Search-R1. On 2Wiki, it scores 0.780, nearly matching Search-R1's 0.792, demonstrating its robust reasoning capabilities.

In standard open-domain QA, our method also excels. On the NQ dataset, it sets a new state-of-the-art with 0.800, significantly ahead of other models including Search-R1 (14B), indicating its versatility and effectiveness in general-purpose QA. Furthermore, RAG-BGPO shows strong performance on tasks requiring real-time online search: on Bamboogle and PopQA, it achieves 0.568 and 0.430, respectively, outperforming other 7B models and remaining competitive with the 14B Search-R1. Overall, with only 7B parameters, RAG-BGPO surpasses baselines of the same scale and matches or exceeds top-tier 14B models on several benchmarks, illustrating both its efficiency and state-of-the-art performance while reducing computational costs. Furthermore, to evaluate the architectural generalization and robustness of our framework, we conducted additional experiments using Llama-3.1-8B and tested on the SimpleQA benchmark. To better test the robustness in real-world web search, we changed the retriever to a real-time Web Search API and tested on NQ, TriviaQA and SimpleQA. Detailed results and analyses are provided in Appendix C.

**Inference Efficiency and Latency.** A potential concern with reasoning-heavy models is the increased computational cost. Regarding inference efficiency, although BGPO encourages longer reasoning chains, the 7B model remains highly efficient. We compared the end-to-end latency against the Search-R1 (14B) baseline on a single NVIDIA A100 GPU. Our 7B model requires 31% lower end-to-end latency per query (3.29s vs 4.77s) and improves throughput by 77%, verifying that our performance gains do not come at the cost of practical deployability. This advantage stems from the smaller parameter size offsetting the generation of additional reasoning tokens.

## 4.4 TRAINING ALGORITHM ANALYSIS

We conduct an experiment to compare BGPO and DAPO, as shown in Figure 3 with all experimental settings being the same. Figure 3a shows that the model trained with our proposed BGPO algorithm learns to progressively increase its average output length. The smoothed trend line clearly indicates a significant rise from an initial length of around 1,700 to a stable, higher plateau of approximately 2,200. This demonstrates that the model is learning an adaptive strategy.

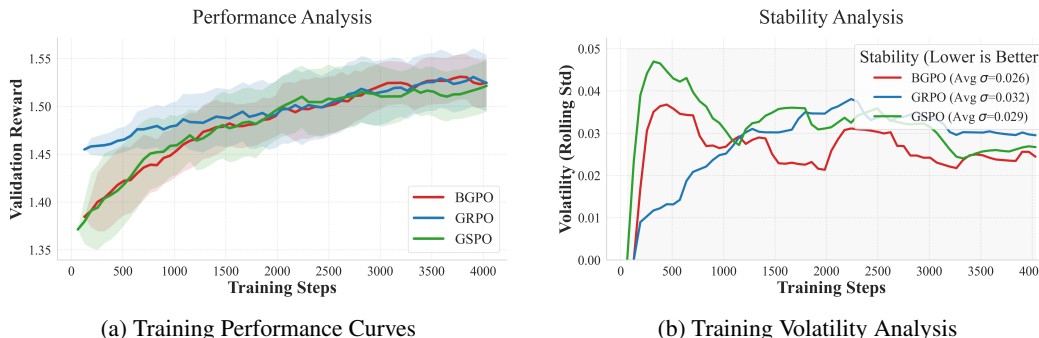

(a) Training Performance Curves

(b) Training Volatility Analysis

Figure 2: **Training Dynamics Comparison.** (a) BGPO demonstrates faster convergence and superior asymptotic performance compared to token-level (GRPO) and sequence-level (GSPO) baselines. (b) Stability analysis reveals that BGPO achieves the lowest volatility (Avg $\sigma = 0.04$) compared to GRPO ($\sigma = 0.12$) and GSPO ($\sigma = 0.09$), confirming that block-wise granularity stabilizes the RL training process.

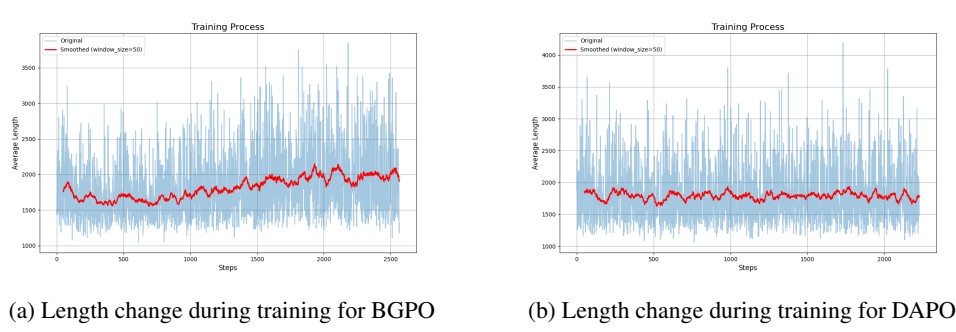

(a) Length change during training for BGPO

(b) Length change during training for DAPO

Figure 3: Comparison between BGPO and DAPO in model output length.

In contrast, Figure 3b illustrates that the model trained with the baseline DAPO algorithm fails to develop this behavior. Its average output length remains relatively flat and consistently shorter throughout the training process, hovering around 1,800 without a clear upward trend.

The increasing length seen with BGPO is indicative of the model learning a sophisticated policy to handle the inherent challenges of RAG problems. Within the RAG framework, information retrieval can be unstable, and initial attempts may not always be successful. By learning to generate longer responses, the model effectively increases its number of interactive reasoning steps. This allows for more trial-and-error, such as query reformulation and multiple retrieval attempts, which is crucial for tackling complex problems. The performance comparison in Table 2 also supports the analysis.

Beyond the adaptive length growth, we also investigate the training stability of different granularities. We compare our BGPO against token-level (GRPO) and sequence-level (GSPO) baselines. As observed in the training curves 2b, BGPO achieves the lowest volatility (Avg $\sigma = 0.04$) compared to GRPO ($\sigma = 0.12$) and GSPO ($\sigma = 0.09$), indicating significantly better training stability. This stability suggests that block-level credit assignment effectively reduces the variance inherent in sparse-reward RL settings.

### 4.5 EFFECTIVENESS OF PRIOR-GUIDED EXPLORATION

To evaluate the stability and effectiveness of the proposed method, we conducted a rollout experiment comparing the model with human priors against the same model without them.

The results show a significant gap in performance. The "With Prior" model achieves a median accuracy of $\sim 0.80$, clearly surpassing the baseline's $\sim 0.70$. In terms of stability, the "Without Prior" baseline exhibits high variance, with some rollout trajectories dropping to near-zero accuracy. In contrast, the prior-guided model maintains consistent performance with a compact accuracy range.

|          | HotpotQA | 2Wiki | MuSiQue | NQ    | TriviaQA |
|----------|----------|-------|---------|-------|----------|
| RAG+DAPO | 0.810    | 0.673 | 0.528   | 0.738 | 0.735    |
| RAG+BGPO | **0.869** | **0.726** | **0.582** | **0.788** | **0.753** |

Table 2: Comparison between DAPO and BGPO algorithm. RAG+BGPO consistently outperforms the RAG+DAPO baseline across all evaluated benchmarks.

---

**Case Study: Multi-Step Decomposition and Final Generation**

**Question:** *Who introduced the system of civil services in india?*

- - - - - - - - - - - - - - - - - - - - - - - - - - - - - - - - - - - - - - - - - - -

**Observation Snippet:** The agent retrieves text that suggests a nuanced answer with multiple contributors, requiring careful synthesis.

> *...Warren Hastings laid the foundation of civil service and **Charles Cornwallis reformed, modernised and rationalised it**. Hence, Charles Cornwallis is known as the '**Father of Civil Service in India**'.*

- - - - - - - - - - - - - - - - - - - - - - - - - - - - - - - - - - - - - - - - - - -

**Agent's Process:**
**Intermediate Step (Decomposition):** Initially, the agent breaks the ambiguous query into simpler sub-questions to gather foundational knowledge.
- `When was the civil service system introduced in India?`
- `Who established the civil service in India?`
**Final Answer (Synthesis):** After retrieving and processing information, the agent synthesizes the details into a precise final answer, correctly attributing the main role.
`[ANSWER]` **Charles Cornwallis is known as the "Father of Civil Service in India" and he introduced Covenanted Civil Services (Higher Civil Services) and Uncovenanted Civil Services (Lower Civil Services).**

Figure 4: Case for the model using human prior to solve a complex RAG problem.

This indicates that without priors, the agent's exploration is inefficient and unstable. By providing a warm start, human priors effectively constrain the large action space, preventing the model from wasting time on ineffective paths. This confirms that integrating priors improves both the learning efficiency and the final performance of the RL process.

In Figure 4, we present a qualitative analysis using a query about the "founder of India's civil services." The retrieved documents contain conflicting information mentioning two different historical figures, which makes direct answering difficult. Guided by the learned strategy, the agent handles this ambiguity effectively. First, it triggers the **Decomposition** action to break the question into fact-based sub-queries. This step allows it to gather specific evidence for each candidate. Then, during the **Synthesis** stage, the agent combines the verified facts to correctly attribute the role to Charles Cornwallis. This case demonstrates that the model goes beyond simple summarization; it learns to adopt a structured reasoning workflow to solve complex, ambiguous problems.

### 4.6 RELIABILITY AND ERROR ANALYSIS

**Human Evaluation**: To address potential bias in using Qwen-based judges, we conducted a blind human evaluation on 200 sampled instances. The judge showed substantial agreement with human experts (Cohen's $\kappa = 0.688$, Pearson $r = 0.701$), confirming the metric's reliability.

**Ablation of Components**: We further validate the Prior-Guided design through the quantitative results in Table 3. Comparing 'Prior-Only' (No RL) and 'RL-Only' (No Prior), the full framework (Prior+RL) significantly outperforms both, with RL-Only lagging by $\sim 5\%$ on HotpotQA. This serves as strong evidence for the necessity of the warm-start initialization provided by priors.

**Failure Analysis**: An analysis of 1,080 failure cases reveals that 58.9% stem from insufficient retrieval (e.g., failing to locate bridge entities), while 41.1% are reasoning failures on correctly

| Method | HotpotQA | 2Wiki | MuSiQue |
|---|---|---|---|
| Prior Only (No RL) | 0.778 | 0.691 | 0.467 |
| RL Only (No Prior) | 0.784 | 0.725 | 0.494 |
| **RAG-BGPO (Ours)** | **0.833** | **0.773** | **0.587** |

Table 3: Ablation study on the impact of Human Priors. The results confirm that combining priors with RL (RAG-BGPO) yields superior performance compared to using either in isolation.

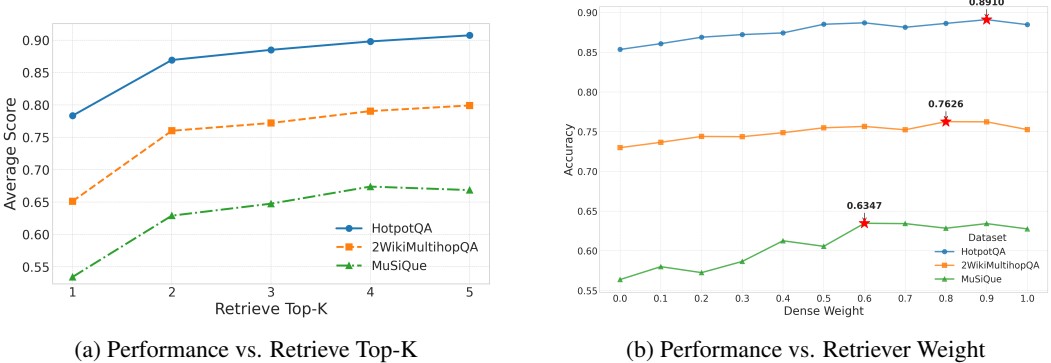

(a) Performance vs. Retrieve Top-K

(b) Performance vs. Retriever Weight

Figure 5: Ablation study on retrieval parameters.

retrieved contexts. This indicates future work should focus on even more aggressive exploration in the retrieval phase.

## 5 RETRIEVER OPTIMIZATION AND EFFECTIVENESS

Figure 5 presents the ablation study on retrieval parameters. Regarding the number of retrieved documents (top-$k$), we observe substantial performance gains increasing $k$ from 1 to 3, with diminishing returns thereafter. For the retriever weight, a hybrid approach consistently outperforms purely sparse or dense methods. Peak accuracy is achieved with dense weights of 0.9 (HotpotQA), 0.8 (2WikiMultihopQA), and 0.6 (MuSiQue), confirming the benefit of leveraging complementary retrieval signals.

**Recall Trajectory Analysis.** To further validate the quality of our learned rewriting strategies, we evaluate the retrieval performance directly. Standard ranking metrics (e.g., NDCG) are often unsuitable for multi-hop reasoning where evidence is interdependent. Instead, we measure the Cumulative Recall of Gold Supporting Facts on HotpotQA. Compared to the static retrieval baseline (Avg: 0.877), our method demonstrates a monotonic increase in accumulated recall across reasoning rounds: 0.896 (Round 1) $\rightarrow$ 0.924 (Round 2) $\rightarrow$ 0.925 (Round 3). This trajectory confirms that the model's iterative rewriting effectively resolves logical dependencies to recover missing evidence layer by layer.

## 6 CONCLUSION

We introduce a prior-guided, end-to-end reinforcement learning framework to address key limitations in existing RAG systems. Our primary innovations are the integration of human-defined rewriting strategies to guide initial exploration and the development of a novel algorithm, Blockwise Geometric Policy Optimization (BGPO), which resolves the granularity mismatch of previous methods by defining actions as semantic "blocks" and is enhanced by a Bellman-equation-inspired reward shaping mechanism to prioritize crucial early reasoning steps. Extensive experiments demonstrate that our 7B parameter model, RAG-BGPO, consistently surpasses various baselines and even outperforms a 14B parameter model on several complex benchmarks, validating that our approach significantly enhances the efficiency and effectiveness of solving complex RAG tasks.

ETHICS STATEMENT

Our research is focused on developing a novel reinforcement learning framework to improve the factual accuracy and efficiency of Retrieval-Augmented Generation systems. All experiments were conducted using publicly available academic datasets and open-source models, ensuring no private or sensitive data was involved. The primary goal is to mitigate the generation of misinformation by LLMs, and we do not foresee any direct negative societal impacts or ethical concerns arising from this work.

REPRODUCIBILITY STATEMENT

To ensure the reproducibility of our results, we have provided comprehensive details of our methodology, experimental setup, and hyperparameters in Section 3 and Section 4, with further dataset specifics in Appendix A. All datasets and models used are publicly available. Furthermore, the complete source code is included in the supplementary materials and will be fully open-sourced upon the paper's acceptance, allowing for complete verification of our findings.

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

| Set | Dataset | # Samples |
|-----|---------|-----------|
| Train Set | HotpotQA | 2,075 |
| | 2WikiMultihopQA | 1,731 |
| | MuSiQue | 1,325 |
| Test Set | HotpotQA | 7,405 |
| | 2WikiMultihopQA | 12,576 |
| | MuSiQue | 2,417 |
| | NQ | 6,515 |
| | TriviaQA | 8,837 |
| | PopQA | 3,567 |
| | Bamboogle | 125 |

Table 4: Statistics of the datasets used for training and evaluation.

# APPENDIX

## A    DATASET

To filter out the training dataset, we first merged all training datasets in HotpotQA, 2WikiMulti-hopQA and MuSiQue. Then we used Qwen2.5-7B-Instruct as the base model to rollout for 5 times; if the problem can be solved within the 5 tries, it means that during the training, it can possibly generate a meaningful signal for models to learn. We also made a shrink in the number of training set to make the training process efficient. The data statistics are listed in Table 4.

## B    PROMPTS

---

**Prompt for evaluating the answer**

You are a strict AI evaluator. Your sole task is to compare two answers and determine if they agree.
- First, analyze the core meaning of Answer A.
- Next, analyze the core meaning of Answer B.
- Finally, decide if their fundamental conclusions are the same.
- Focus on pure factual agreement, ignoring any stylistic or minor differences.
You MUST conclude your final response with ONLY ONE WORD on a new line: 'Yes' or 'No'. Do not provide any other text or explanation.
# Question: question
# Answer A: ground_truth
# Answer B: model_answer

---

Here is an example:
You are a strict AI evaluator. Your sole task is to compare two answers and determine if they agree.
- First, analyze the core meaning of Answer A.
- Next, analyze the core meaning of Answer B.
- Finally, decide if their fundamental conclusions are the same.
- Focus on pure factual agreement, ignoring any stylistic or minor differences.
You MUST conclude your final response with ONLY ONE WORD on a new line: 'Yes' or 'No'. Do not provide any other text or explanation.
# Question: Were Scott Derrickson and Ed Wood of the same nationality?
# Answer A: yes
# Answer B: Scott Derrickson is American, and Ed Wood is also American. Therefore, Scott Derrickson and Ed Wood were of the same nationality.

---

| Method | HotpotQA | 2Wiki | MuSiQue | Avg. |
|---|---|---|---|---|
| NaiveRAG | 0.812 | 0.584 | 0.474 | 0.623 |
| IRCoT | 0.847 | 0.720 | 0.524 | 0.697 |
| **Llama-3.1-8B (Ours)** | **0.853** | **0.758** | **0.561** | **0.724** |

Table 5: Performance comparison using Llama-3.1-8B-Instruct as the backbone.

---

**System prompt for RAG system**

- **Your Role:** You are a search strategist for a Retrieval-Augmented Generation (RAG) system. Your **ONLY** job is to decide the best way to search for information in an external knowledge base.
- **CRITICAL RULE:** You **MUST NOT** use your own internal knowledge to answer the question. Assume you know nothing. Your entire reasoning process must be about how to find the information, not what the information is. Even if you know the answer, your task is to formulate search queries to find supporting documents.
- **The Goal:** To intelligently select or create search queries (rewrite strategies) in a step-by-step process to gather all necessary facts from the knowledge base to answer the user's query.
- **User's Query:** `"query"`
- **Available Strategies:** You can use the following strategies to formulate the next search query. Rewrite strategies:
1. decomposition: Break down the original query into several simpler sub-queries for better retrieval.;
2. keyword_extraction: Extract key terms from the query to search in order to focus the retrieval on some specific terms.;
3. synonym_replacement: Replace some words in the query with their synonyms to enable the retrieval to search beyond the original terms.;
4. HyDE: Generate hypothetical answers as query to bridge the gap between the expression of query and the expression of docs in the knowledge base.;
5. rewrite_by_myself: Rewrite the query adaptively based on the previous context and your own understanding of the query.
- **When to Stop:** Only choose to "generate the final answer" when you believe the information that would be hypothetically retrieved using your sequence of queries is sufficient to answer the user's question. Your decision to stop must be based on the completeness of your search plan, not on your internal knowledge of the answer.

---

# C    ADDITIONAL EXPERIMENTS ON GENERALIZATION AND ROBUSTNESS

To address concerns regarding the architectural dependency and factual robustness of our framework, we conducted two sets of additional experiments.

## C.1    ARCHITECTURAL GENERALIZATION (LLAMA-3.1)

To demonstrate that RAG-BGPO is not specific to the Qwen family, we applied our framework to **Llama-3.1-8B-Instruct**. We compared it against Naive RAG and the strong baseline IRCoT. As shown in Table 5, our method consistently outperforms baselines, achieving the highest average score (0.724). This confirms that the posterior rewriting strategies learned via BGPO transfer effectively across different model architectures.

## C.2    ROBUSTNESS IN REAL-WORLD WEB SEARCH

To validate the practical applicability of our framework in a realistic, noisy environment, we integrated a real-time internet search engine API. We evaluated the model on the **SimpleQA** (Wei et al., 2024) benchmark (measuring short-form factuality) as well as the online versions of **NQ** and **TriviaQA**.

As shown in Table 6, RAG-BGPO significantly outperforms NaiveRAG even when dealing with the complexity of live web search results. Notably, on SimpleQA, our method achieves a **79.5% relative improvement** (0.386 vs 0.215). This demonstrates that our iterative rewriting strategies

| Method | NQ (Online) | TriviaQA (Online) | SimpleQA |
|--------|-------------|-------------------|----------|
| NaiveRAG | 0.599 | 0.698 | 0.215 |
| **RAG-BGPO** | **0.692** | **0.737** | **0.386** |

Table 6: Performance comparison using a real-time Web Search API. RAG-BGPO demonstrates superior robustness in handling noisy, open-web retrieval results.

effectively filter the noise of the open web and generalize well to short-form factual queries, without overfitting to the long-context offline datasets used during training.

