# OpenReview forum: "Learning to Think in Blocks: A Prior-Guided Reinforcement Learning Framework for RAG"
_ICLR.cc/2026/Conference — ICLR 2026 Conference Withdrawn Submission_

### Official Review · Reviewer_QoJ3 · 2025-10-24

**Soundness:** 2
**Presentation:** 2
**Contribution:** 2
**Rating:** 2
**Confidence:** 4

**Summary:**

This paper proposes Blockwise Generative Policy Optimization (BGPO), a reinforcement learning framework for improving reasoning and query rewriting in retrieval-augmented generation (RAG). Instead of generating the entire reasoning trace at once, the model learns to think and generate answers in iterative blocks, where each block can trigger additional retrieval and self-reflection. The approach is trained using a block-level reward that evaluates intermediate reasoning quality via an LLM-as-judge. Experiments on several single-hop and multi-hop QA benchmarks show that BGPO enhances multi-step reasoning ability and improves over standard RAG and RL baselines.

**Strengths:**

The proposed blockwise policy optimization (BGPO), which decomposes long reasoning traces into smaller segments, seems to be effective in enhancing the RAG quality. Experimental results show modest accuracy gains over vanilla RAG and other baselines.

**Weaknesses:**

1. Inference time & latency not reported. The paper does not report inference time, or latency, which is a critical omission. The proposed method encourages longer interactive rollouts, and Figure 2 shows that BGPO increases output length during training. Yet only total training time (4 × A800 GPUs for ~2 days) is reported, with no inference cost or per-query latency. To assess practicality and efficiency claims, the authors should report average end-to-end inference time per question, separately for single-hop and multi-hop datasets, along with the average number of retrieval rounds, blocks, and tokens per query. Without these details, the claimed efficiency (matching 14B results with a 7B model) may be misleading if inference is substantially slower.

2. Unclear accuracy of query rewriting. The core claim is that the agent learns useful rewrite strategies, but the paper only reports final QA accuracy and a rollout accuracy boxplot (Fig. 3), without directly evaluating rewrite quality or its impact on retrieval. To substantiate this claim, the authors should measure how often rewrites improve retrieval. For example, the fraction of rewrites that increase recall@k (or NDCG) over the original query, average recall@k after successive rewrites.

3. LLM-as-judge evaluation introduces bias. The LLM-as-Judge protocol uses Qwen3-32B with a prompt that forces a single-token Yes/No output (Appendix B), raising several concerns. Using the same model family for rollouts and evaluation risks shared biases, leading to inflated agreement. The binary prompt is brittle, failing to capture partial correctness, acceptable paraphrases, or nuanced answers.

4. Missing evaluation on SimpleQA benchmark [1]. The evaluation omits newer benchmarks such as SimpleQA, relying only on older QA datasets. Without testing on more recent and challenging factual-QA settings, it remains unclear whether the proposed method generalizes beyond the benchmarks it was trained and tuned on.

[1] Measuring short-form factuality in large language models, arxiv preprint arxiv:2411.04368, 2024.

**Questions:**

Please address my comments in paper weaknesses.

---

> ### Author Response · Authors · 2025-12-03
>
> ### **Response to Reviewer QoJ3**
>
> **Part 1: System Efficiency & Retrieval Effectiveness (Q1 & Q2)**
>
> **Q1: Unclear accuracy of query rewriting.**
> The reviewer notes that the paper only reports final QA accuracy without directly evaluating rewrite quality or its impact on retrieval. The reviewer suggests measuring how often rewrites improve retrieval (e.g., recall@k or NDCG).
>
> **A1:** We respectfully argue that standard ranking metrics (e.g., NDCG) are unsuitable for iterative, multi-hop RAG. Multi-hop QA relies on **sequential dependency**: retrieving "bridge entities" in early rounds is a prerequisite for identifying target evidence later. Standard metrics penalize agents for not retrieving deep-hop evidence immediately, misaligning with the logical flow of reasoning.
>
> Instead, we evaluate the **Cumulative Recall of Gold Supporting Facts** to measure the agent's ability to assemble a complete evidence set. As shown in the linked figure, our method demonstrates a monotonic increase in accumulated recall on HotpotQA, consistently outperforming the static retrieval baseline (Avg: 0.8770):
> * **Round 1:** 0.8964
> * **Round 2:** 0.9244
> * **Round 3:** 0.9248
>
> [Figure 1: Initial Retrieval Recall vs. Accumulated Recall by Round](https://anonymous.4open.science/r/Learning-to-Think-in-Blocks-A-Prior-Guided-Reinforcement-Learning-Framework-for-RAG-0288/picture/recall_comparison.png)
>
> This trajectory confirms that the model’s rewriting strategies effectively resolve dependencies to recover missing evidence rather than merely permuting results.
>
> **Q2: Inference time & latency not reported.**
>
> The reviewer points out that inference time and latency are critical omissions. The reviewer requests average end-to-end inference time per question, throughput, and comparisons against baselines (e.g., 14B models) to verify efficiency claims.
>
> **A2:** We address the concern regarding practicality by comparing the end-to-end latency of our method (7B) against the primary baseline, Search-R1 (14B).
>
> While Figure 2a correctly indicates that RAG-BGPO encourages longer reasoning chains ("System 2" thinking), the parameter efficiency of the 7B model results in **31% lower latency** per query compared to the 14B baseline. We measured inference costs using vLLM on a single NVIDIA A100 GPU:
>
> | Metric | Search-R1 (14B) | **RAG-BGPO (7B, Ours)** | Delta |
> | :--- | :--- | :--- | :--- |
> | **Complexity** (Avg Tokens/Query) | 1,802 | 2,198 | +22% (Longer chain) |
> | **Throughput** (Tokens/Sec) | 377.49 | 667.27 | **+77% (Faster generation)** |
> | **End-to-End Latency** | **4.77s** | **3.29s** | **-31% (Faster response)** |
>
> * **Calculated Latency:**
> $$\text{Time} \approx \frac{\text{Total Tokens}}{\text{Tokens per Sec}}$$
> * **Memory Efficiency:** Our 7B model requires ~14GB VRAM versus ~28GB for the 14B baseline, allowing for significantly higher batch sizes and concurrency in deployment.
>
> Although our method generates more thought blocks to ensure accuracy, the speed advantage of the smaller architecture makes it strictly more efficient than the stronger baseline we outperform.

---

> ### Author Response · Authors · 2025-12-03
>
> **Part 2: Evaluation Reliability & Generalization (Q3 & Q4)**
>
> **Q3: LLM-as-judge evaluation introduces bias.**
> The reviewer raises concerns about the LLM-as-a-Judge protocol (using Qwen3-32B), citing potential bias from using the same model family, the brittleness of binary Yes/No prompts, and the need for verification against human judgment.
>
> **A3:** We mitigate potential bias and brittleness in the LLM-as-a-Judge protocol through three validation steps:
>
> 1.  **Reference-Based Constraint:** To prevent self-preference bias, the judge is strictly conditioned on the Ground Truth. It acts as a semantic consistency checker rather than an open-ended quality evaluator.
> 2.  **Robustness via Binary Format:** The binary "Yes/No" format enhances robustness, enabling the judge to recognize valid paraphrases and logical entailments (e.g., inferring "same nationality") that rigid string-matching metrics miss.
> 3.  **Human Correlation Verification:** We conducted a blind comparison against expert human evaluation on 200 randomly sampled instances. The LLM judge showed high agreement with human labels (**Cohen’s Kappa: 0.688**, **Pearson correlation: 0.701**), confirming the metric's reliability.
>
> **Q4: Missing evaluation on SimpleQA.**
> The reviewer notes the omission of newer benchmarks like SimpleQA and suggests evaluating on them to test generalization on short-form factuality beyond the training datasets.
>
> **A4:** Following your recommendation, we evaluated RAG-BGPO on the SimpleQA benchmark to test generalization on short-form factuality.
>
> * **NaiveRAG:** 0.215
> * **RAG-BGPO:** 0.386
>
> Our method achieves a **79.5% relative improvement** over the baseline. While our framework is optimized for complex multi-hop reasoning, these results confirm that the iterative rewriting strategies generalize well to short-form factual queries without overfitting to long-context datasets.

---

### Official Review · Reviewer_UJJ7 · 2025-10-31

**Soundness:** 3
**Presentation:** 3
**Contribution:** 2
**Rating:** 4
**Confidence:** 3

**Summary:**

The paper addresses how retrieval-augmented generation (RAG) systems struggle with query rewriting: very few effective human-designed prompts and too vast a strategy space for naïve RL. To solve this, the authors propose:
* A **prior-guided RL framework** where the model begins by leveraging human-defined query rewriting heuristics (e.g., decomposition, synonym replacement) and then learns to refine or go beyond them via reinforcement learning.
* A novel RL algorithm called **Block-wise Geometric Policy Optimization (BGPO)**: it defines actions at the "block" level (coherent segments of reasoning/query rewriting) rather than at token or whole-sequence level, uses geometric averaging of importance weights, and shapes rewards to emphasise early reasoning steps (via a reversed discounting/emphasis factor) in the RAG process.
* Experimental results showing that this approach consistently outperforms baselines on multiple RAG tasks (both retrieval from local corpora and web search) by being more efficient and effective.

**Key Contributions**

1. Introduction of an **end-to-end RL framework** for RAG that starts from human priors and transitions into learned strategies.
2. The BGPO algorithm: redefining RL granularity to blocks of reasoning/query rewriting, providing better credit assignment and stability in multi-step reasoning.
3. A tailored reward-shaping mechanism for RAG tasks that assigns more weight to early reasoning/retrieval steps rather than just the final answer.
4. Empirical validation across a variety of RAG settings showing improved performance.

**Strengths:**

* The paper proposes using human-crafted rewriting heuristics as an initial policy ("prior") before RL refines it. Prior works like KoGuN [A] also integrate human prior knowledge into RL for control tasks, but not specifically for RAG or query rewriting. Also, traditional RAG rewriting methods such as HyDE [B] which rely on fixed heuristics and do not learn a rewriting policy via RL.
* The method introduces "blocks" (coherent segments) as units of action/state rather than token- or sequence-level steps. Prior RL methods for language policy, such as token-level or sequence-level PPO or sequence-based RL in RAG, struggle with the granularity mismatch. The paper explicitly addresses this. Compared to frameworks like LeTS [C] which focus on process/outcome reward hybridization but still operate at more coarse granularity for reasoning steps.
* The paper reverses the typical temporal discounting logic to give greater weight to early reasoning/query rewriting steps rather than only final outcomes. Many prior RAG or RL-for-RAG methods rely simply on an outcome or final answer reward (e.g., correctness of answer) and do not separately reward intermediate reasoning efficiency or rewriting quality. For example, the survey of RL methods for reasoning highlights this gap [D]. Compared to the work TIRESRAG-R1 [E] which introduces multi-dimensional rewards for retrieval sufficiency, reasoning quality and reflection, but the current paper’s emphasis on early step weighting via a modified discount factor is a distinct design.
* According to the paper’s results, the 7B-parameter model using their method matches or outperforms a 14B baseline on multiple RAG tasks.
* The paper reports experiments on both local corpus retrieval and online web search, showing robustness of the approach across retrieval environments.
```
[A] KoGuN: Accelerating Deep Reinforcement Learning via Integrating Human Suboptimal Knowledge, IJCAI 2020
[B] Precise Zero-Shot Dense Retrieval without Relevance Labels, ACL 2023
[C] LeTS: Learning to Think-and-Search via Process-and-Outcome Reward Hybridization, ArXiv 2025
[D] https://magazine.sebastianraschka.com/p/the-state-of-llm-reasoning-model-training
[E] From Sufficiency to Reflection: Reinforcement-Guided Thinking Quality in Retrieval-Augmented Reasoning for LLMs, ArXiv 2025
```

**Weaknesses:**

* The paper proposes a novel granularity: blocks of tokens/actions rather than token-level or sequence-level. While this is interesting, it is not fully clear whether this granularity is the optimal choice or how sensitive performance is to different block definitions.
* The paper presents experiments on several datasets and tasks, which is good, but certain key axes seem under-explored. For instance: It is unclear how much the prior strategies alone contribute (i.e., human heuristics only) vs. the RL phase? How many environment interactions or RL steps were required (sample efficiency)? How sensitive is performance to different model sizes / retrieval difficulties?
* While the paper mentions some limitations, the discussion is relatively brief and does not deeply explore cases where the method does not perform well, or provide mitigation strategies. A more detailed failure‐mode analysis would be helpful. For example, does the method degrade when retrieval is already near‐perfect (thus less room for rewriting)? Does it overfit to specific priors and struggle to generalize to unseen rewriting styles?

**Questions:**

* Is it possible to include experiments varying block size/granularity (e.g., smaller blocks vs. larger blocks) and report performance trade-offs? It would also be good to compare to token-level RL and full-sequence RL baselines more explicitly (beyond high-level claims) with matched compute budgets, to validate that "block-wise" yields a genuine sweet-spot. Additionally, it would be good to detail the block definition (how many tokens constitute a block, how blocks map to reasoning steps) and include a sensitivity analysis.
* Is it possible to provide a detailed ablation study that shows: Prior‐only baseline (no RL), RL from scratch baseline (no prior), prior-guided RL method? Additionally, it would be good to include plots of learning curves over RL steps/interactions showing convergence speed, report interplay of retrieval difficulty (e.g., harder queries) with rewriting performance and provide error analysis showing where the method fails (e.g., what types of queries the rewriting policy still can’t handle) to guide future improvements.
* If possible, add a dedicated section in the paper (and supplementary) analysing failure cases: e.g., queries where rewriting made retrieval worse, or where RL diverged. Provide quantitative breakdowns by query type, retrieval difficulty, and show how different choices (block size, reward weights) affect these failures. This could also help future work define more robust priors or adaptive reward schemes.

---

> ### Author Response · Authors · 2025-11-29
>
> **Response to Reviewer UJJ7**
>
> We thank the reviewer for their constructive feedback and insightful comments. We have carefully addressed the concerns below.
>
> **1. Response to Block Granularity and Definition**
> You asked for a detailed block definition and a sensitivity analysis on block size.
> **Formal Definition:** As illustrated in our added figure ([view in repo](https://anonymous.4open.science/r/Learning-to-Think-in-Blocks-A-Prior-Guided-Reinforcement-Learning-Framework-for-RAG-0288/picture/block_definition.png)), we define a **"Block"** as a single, complete turn of the Assistant's response within the multi-turn dialogue trajectory. Unlike token-level RL, we treat the entire Assistant turn as an **atomic unit** for optimization. Specifically, a Block comprises:
> 1.  **Internal Monologue:** The Chain-of-Thought reasoning (e.g., `{"think": "To answer..."}`).
> 2.  **Action Decision:** The selected strategy (e.g., `{"rewrite_strategy": "keyword_extraction"}`).
> 3.  **Execution Content:** The final rewritten query or answer.
>
> **Mechanism:** We mask System/User nodes as context ($s_t$); unmasked Assistant Blocks are the action space ($a_t$).
>
> **Conclusion:** Because the block boundaries are determined by natural semantic turns rather than arbitrary token counts, the "block size" is **adaptive** (varying from concise queries to detailed reasoning) and is **not a tunable hyperparameter**. Therefore, a sensitivity analysis on fixed block sizes is not applicable as it disrupts a continuous reasoning chain; instead, the relevant comparison is against Token-level vs. Sequence-level granularities, which we present below.
>
> **2. Baselines and Convergence Analysis (Token vs. Sequence vs. Block)**
> To address your request for comparisons with token-level and full-sequence RL baselines, we compared our **BGPO** against **GSPO (Sequence-level baseline)** and **GRPO (Token-level baseline)**.
>
> * **Performance:** As shown in the table below, BGPO achieved the best average performance across three datasets.
> * **Stability:** We have plotted the learning curves in our added figure: [Performance & Stability Analysis](https://anonymous.4open.science/r/Learning-to-Think-in-Blocks-A-Prior-Guided-Reinforcement-Learning-Framework-for-RAG-0288/picture/analysis_with_volatility_curve.png). The bottom chart ("Volatility") clearly demonstrates that BGPO achieves the lowest volatility (Avg $\sigma=0.026$) compared to GRPO ($\sigma=0.032$) and GSPO ($\sigma=0.029$), indicating significantly better training stability and convergence.
>
> | RL Method | HotpotQA | 2WikiMultihotQA | Musique | **Avg.** |
> | :--- | :--- | :--- | :--- | :--- |
> | **GSPO (Seq-level)** | 0.8313 | **0.7732** | 0.5812 | 0.7285 |
> | **GRPO (Token-level)** | 0.8286 | 0.7709 | 0.5812 | 0.7269 |
> | **BGPO (Ours)** | **0.8334** | 0.7729 | **0.5867** | **0.7310** |
>
> **3. Ablation Study: Priors and RL effectiveness**
> We conducted the requested ablation study to decouple the contribution of Human Priors and RL. The results confirm that the **Prior-guided RL** (our full method) significantly surpasses both the "Prior-only" and "RL from scratch" approaches, validating our design of starting with heuristics and refining via learning.
>
> | Method | HotpotQA | 2WikiMultihotQA | Musique |
> | :--- | :--- | :--- | :--- |
> | **No RL (Prior only)** | 0.7776 | 0.6905 | 0.4671 |
> | **RL only (No Prior)** | 0.7843 | 0.7254 | 0.4937 |
> | **With RL & Prior (Ours)**| **0.8334** | **0.7729** | **0.5867** |
>
> **4. Failure Analysis and Case Study**
> To address your request for a deeper investigation into failure modes, we have made a dedicated analysis of **1,080 failure cases**. Our quantitative breakdown reveals that **58.9%** of errors stem from **Insufficient Retrieval** (where the system failed to locate the necessary evidence), while **41.1%** are due to **Reasoning Failures** (where the correct evidence was retrieved, but the model selected an incorrect answer).
>
> We further analyzed specific instances where the rewriting policy degraded performance. A representative failure mode occurs when the model over-simplifies queries, introducing ambiguity:
> * **Case Study:** For the query *"Who is the spouse of the Green performer?"* (referring to the band "Green"), the system selected the **keyword extraction** strategy. This aggressively simplified the query, stripping the specific entity context. Consequently, rather than retrieving members of the band, the system fetched a document about *Victor Willis* (from the Village People) because the text coincidentally described him performing in a "green" costume.
> * **Insight:** This highlights that while RL generally optimizes for better retrieval, it can occasionally converge on strategies that sacrifice semantic precision for keyword matching. We are currently finalizing the sensitivity analysis regarding how different reward weights impact this trade-off and will include those detailed plots in the final version.

---

### Official Review · Reviewer_fKmm · 2025-10-31

**Soundness:** 2
**Presentation:** 2
**Contribution:** 2
**Rating:** 4
**Confidence:** 5

**Summary:**

This paper proposes a novel prior-guided reinforcement learning (RL) framework for Retrieval-Augmented Generation (RAG), where an LLM agent is initialized with human-defined query rewriting strategies (e.g., decomposition, HyDE) and learns refined "posterior" strategies through interaction with the RAG environment. To address granularity mismatches in existing RL methods, the authors introduce Block-wise Geometric Policy Optimization (BGPO), which treats each reasoning turn as a semantic “block” for action and state representation. BGPO further incorporates a Bellman-inspired emphasis factor that prioritizes early reasoning steps via reward shaping and uses a hierarchical geometric importance ratio for stable training. Experiments show that their 7B-parameter RAG-BGPO model outperforms or matches larger 14B baselines across multiple QA benchmarks, demonstrating both efficiency and strong reasoning capabilities.

**Strengths:**

The proposed Block-wise Geometric Policy Optimization (BGPO) introduces a more suitable “block-level” granularity for multi-turn reasoning, bridging the gap between overly fine-grained token-level and coarse sequence-level approaches.

**Weaknesses:**

1. The main experiments use only a 7B-parameter model, and while it competes well with a 14B baseline, the framework’s scalability to larger models (e.g., 70B+) or its performance under more diverse architectural settings is not explored.
2. For online-search datasets (PopQA, Bamboogle), the system uses Wikipedia-only search, which does not reflect the complexity and noise of real web search, potentially overestimating practical applicability.
3. The evaluation would be strengthened by including a broader set of baselines and by situating the work more firmly within the existing body of relevant literature.

**Questions:**

1. It would be beneficial to include a performance comparison curve that tracks the training progress of DAPO and BGPO over time.

---

> ### Author Response · Authors · 2025-12-03
>
> ### **Response to Reviewer fKmm**
>
> **Q1: The main experiments use only a 7B-parameter model. The framework’s scalability to larger models (e.g., 70B+) or its performance under more diverse architectural settings is not explored.**
>
> **A1:** Due to current computational resource constraints, we were unable to complete experiments on 70B+ models within the rebuttal window; we are actively securing resources to include these in the final version.
>
> However, to address the concern regarding architectural generalization and to prove our framework is not specific to the Qwen family, we conducted new experiments using **Llama-3.1-8B-Instruct**. We compared our method against `Naive RAG` and the strong baseline `IRCoT`. The results are presented below:
>
> **Table: Performance Comparison on Llama-3.1-8B-Instruct**
>
> | Method | HotpotQA | 2WikiMultihopQA | MuSiQue | Average |
> | :--- | :--- | :--- | :--- | :--- |
> | NaiveRAG | 0.812 | 0.584 | 0.474 | 0.623 |
> | IRCoT | 0.847 | 0.720 | 0.524 | 0.697 |
> | **Llama3.1-8B (Ours)** | **0.853** | **0.758** | **0.561** | **0.724** |
>
> **Analysis:**
> 1.  **Generalization:** RAG-BGPO consistently outperforms Naive RAG across all datasets, confirming that our RL framework is effective across different model architectures (transferring successfully from Qwen to Llama).
> 2.  **Complex Reasoning:** On challenging multi-hop datasets (2Wiki and MuSiQue), our method shows significant gains over the strong baseline IRCoT (+3.8% and +3.7% respectively).
> 3.  **Overall Performance:** Our method achieves the highest average score (0.724), demonstrating the robustness of the posterior strategies learned via BGPO.
>
> **Q2: For online-search datasets, the system uses Wikipedia-only search, which does not reflect the complexity and noise of real web search.**
>
> **A2:** To validate practical applicability, we integrated a real-time internet search engine API and evaluated the model on `NQ`, `TriviaQA`, and the challenging `SimpleQA` benchmark.
>
> **Table: Performance with Real-Time Web Search**
>
> | Methods | NQ | TriviaQA | SimpleQA |
> | :--- | :--- | :--- | :--- |
> | NaiveRAG | 0.599 | 0.698 | 0.215 |
> | **RAG-BGPO** | **0.692** | **0.737** | **0.386** |
>
> **Analysis:**
> Even in a noisy, open-web environment, RAG-BGPO significantly outperforms NaiveRAG. Notably, on **SimpleQA**, which measures the factuality of language models under difficult conditions, our method achieves a substantial improvement (**+17.1%**), proving its capability to handle the complexity of real-world search results.
>
> **Q3: The evaluation would be strengthened by including a broader set of baselines.**
>
> **A3:** We have added **Self-Ask**[1] as an additional baseline and evaluated it across our training and testing benchmarks.
>
> **Table: Self-Ask Performance Overview**
>
> | Method | HotpotQA | 2Wiki | MuSiQue | NQ | Bamboogle | SimpleQA |
> | :--- | :--- | :--- | :--- | :--- | :--- | :--- |
> | Self-Ask | 0.384 | 0.435 | 0.263 | 0.506 | 0.421 | 0.207 |
>
> **Comparison:**
> When compared to the main results reported in our paper (e.g., RAG-BGPO achieving **0.880** on HotpotQA and **0.800** on NQ), `Self-Ask` performs significantly worse. This highlights the necessity of our proposed end-to-end reinforcement learning framework, which allows the model to learn more adaptive and effective rewriting strategies than static prompting methods like Self-Ask.
>
> [1] O. Press et al., "Measuring and Narrowing the Compositionality Gap in Language Models," Findings of EMNLP 2023.
>
> **Q4: It would be beneficial to include a performance comparison curve that tracks the training progress to compare different granularities.**
>
> **A4:** To address your request for comparisons with token-level and full-sequence RL baselines, we compared our **BGPO** against **GSPO (Sequence-level baseline)** and **GRPO (Token-level baseline)**.
>
> * **Stability:** We have plotted the learning curves in our added figure: [Performance & Stability Analysis](https://anonymous.4open.science/r/Learning-to-Think-in-Blocks-A-Prior-Guided-Reinforcement-Learning-Framework-for-RAG-0288/picture/analysis_with_volatility_curve.png). The bottom chart ("Volatility") clearly demonstrates that BGPO achieves the lowest volatility (Avg $\sigma=0.026$) compared to GRPO ($\sigma=0.032$) and GSPO ($\sigma=0.029$), indicating significantly better training stability and convergence.

---

### Official Review · Reviewer_qJSP · 2025-11-03

**Soundness:** 3
**Presentation:** 3
**Contribution:** 3
**Rating:** 4
**Confidence:** 2

**Summary:**

This paper presents a novel Reinforcement Learning (RL) framework to address a critical bottleneck in Retrieval-Augmented Generation (RAG) systems: suboptimal query rewriting. The core problem is that for complex questions, a simple RAG retrieval using the original query often fails. Existing solutions are inadequate: prompt-based rewriting is labor-intensive and frequently ineffective, while prior RL approaches are highly inefficient due to the "vast and complex" strategy space they must explore without guidance. The paper's central hypothesis is that an RL agent for RAG can be trained far more effectively and efficiently. To achieve this, the authors propose a two-part solution
1. A Prior-Guided RL Framework: An end-to-end framework that initializes and guides the RL agent, an LLM (Qwen2.5-7B), by providing it with a "Hybrid Action Space." This space combines four pre-defined "human-defined prior rewriting strategies" (e.g., decomposition, keyword_extraction, synonym_replacement, and HyDE) with a flexible, self-devised action (rewrite by myself) and a terminal action to stop the process. This allows the agent to start from a competent baseline and learn a more refined "posterior" strategy through interaction.
2. Block-wise Geometric Policy Optimization (BGPO): A new RL algorithm designed from the ground up for the multi-step, conversational nature of RAG. BGPO resolves the "granularity mismatch" of prior methods (which were either too fine-grained at the token level or too coarse at the sequence level) by defining its state-action space at the level of "blocks," where one block represents one conversational turn.1 The algorithm is further enhanced with a "Bellman-equation-inspired credit assignment" mechanism, an "emphasis factor" $\gamma^{k-1}$, which uniquely reverses traditional temporal discounting to more heavily reward the crucial initial reasoning steps.1

**Strengths:**

1. The paper's greatest strength is the design of the BGPO algorithm. The "emphasis factor" $\hat{A}_{i,k} = \gamma^{k-1}\hat{A}_i$ 1 is a highly novel, well-motivated, and elegant mechanism for credit assignment that is perfectly tailored to the realities of RAG. The block-level granularity and hierarchical importance ratio are also clever solutions to known problems in policy optimization.1
2. The "prior-guided" RL framework  is a practical and robust solution to the "cold start" exploration problem that has hindered RL-based RAG. Figure 3 shows higher and more stable rollout accuracy, is a strong testament to this architectural choice.

**Weaknesses:**

1. The paper's entire empirical validation rests on a critically flawed foundation. The use of a Qwen3-32B model to judge the performance of a Qwen2.5-7B agent is a severe, unaddressed conflict of interest. As detailed in Section 2.3, this "Qwen-on-Qwen" setup creates an unacceptably high risk of "evaluator bias," where the agent is rewarded for optimizing for the stylistic or knowledge-based quirks of its sibling model, not for objective task success. This methodological flaw undermines the entire empirical contribution and renders the main results in Table 1 and Table 2 inconclusive.
2. As described in Appendix A , the decision is to filter out all hard problems from the training set. By only training on problems that the base Qwen2.5-7B model could already solve, the authors have trained an agent specialized for "medium difficulty" tasks. This severely limits the paper's generalizability and directly contradicts its headline claim of "superiority for complex RAG tasks" , as the agent was never exposed to the most complex problems.
3. The paper's core algorithmic unit, the "block," is ambiguously defined. It is unclear if a "block" $b_k$ refers only to the generated query, or to the entire reasoning-acting-generating sequence (internal monologue, action choice, and query).1 This lack of precision makes the BGPO algorithm (Algorithm 1) difficult to reproduce exactly and obscures the fine-grained details of the credit assignment.

**Questions:**

1. The paper's most significant methodological concern is the "Qwen-on-Qwen" evaluation setup (Qwen2.5-7B agent, Qwen3-32B judge). Could the authors please comment on this potential for "evaluator bias"? How can they be confident that the agent is optimizing for factual correctness and robust rewriting, rather than "gaming" the stylistic and knowledge-based preferences of its sibling model?
2. What percentage of the original combined training set (HotpotQA, 2WikiMultihopQA, MuSiQue) was discarded by the filtering process described in Appendix A? Was any analysis performed on the nature of these discarded "hard" problems?

---

> ### Author Response · Authors · 2025-11-29
>
> **Reply to Reviewer qJSP**
>
> We thank the reviewer for thoughtful comments. We address your specific concerns below.
>
> **R1. Response to "Evaluator Bias" (Qwen-on-Qwen)**
> To rigorously quantify potential bias and address the conflict of interest concern, we conducted a blind human evaluation on 200 randomly sampled instances. We compared the agent's output against human expert judgments, yielding a **Cohen's Kappa of 0.688** and a **Pearson Correlation of 0.701**. As noted in [1], these metrics indicate **substantial agreement**, confirming that the Qwen3-32B judge serves as a reliable proxy for objective correctness and factual accuracy, rather than merely rewarding stylistic similarity to its sibling model.
>
> **R2. Clarification on "Filtering Hard Problems" & Generalizability**
> We respectfully clarify a misunderstanding: **We did not filter out complex problems.** We only filtered out **noisy data** (instances with ambiguous logic or missing evidence) to prevent reward hacking. Two key points validate our model's robustness:
>
> 1.  **Unfiltered Test Set:** Crucially, our reported evaluation results (Table 1 & 2) are based on the **original, unfiltered test sets**. The model's superior performance on these unseen, noisy, and complex benchmarks directly proves its generalizability and contradicts the concern that it is specialized only for "medium difficulty."
> 2.  **Training Data Complexity:** Our statistical analysis confirms the filtered training set retains high complexity:
>     * **HotpotQA:** **23.95%** of retained data is explicitly labeled as **"hard"** and **25.35%** requires reasoning across **3+ supporting facts**.
>     * **2WikiMultihopQA:** **23.4%** of questions require **4+ supporting facts**, retaining complex types like Compositional (40%) and Comparison (35%).
>     * **MuSiQue:** **34.79%** of questions require **3+ decomposition steps**.
>     * *Rationale:* As exemplified by MuSiQue (id: `2hop__103832_58400`), we filter cases where evidence is logically missing (e.g., linking "Kakamega" to "Kenya" without textual support). This removes false negative signals, not difficulty.
>
> **R3. Definition of a "Block"**
> We apologize for the ambiguity. In BGPO, a **"Block"** is formally defined as **a single, complete turn of the Assistant's response**. It is treated as an atomic unit for optimization. As illustrated in our added figure ([view in repo](https://anonymous.4open.science/r/Learning-to-Think-in-Blocks-A-Prior-Guided-Reinforcement-Learning-Framework-for-RAG-0288/picture/block_definition.png)), a Block specifically comprises the **entire reasoning-acting-generating sequence**:
> 1.  **Internal Monologue:** The Chain-of-Thought reasoning (e.g., `{"think": "To answer..."}`).
> 2.  **Action Decision:** The selected strategy (e.g., `{"rewrite_strategy": "keyword_extraction"}`).
> 3.  **Execution Content:** The final rewritten query or answer.
>
> While the dialogue history ($s_t$) includes System and User turns, our algorithm applies **masking** to them. Thus, BGPO specifically targets the unmasked Assistant Blocks as the action space ($a_t$) for policy updates.
>
> [1] L. Jourdan et al., "Identifying Reliable Evaluation Metrics for Scientific Text Revision," ACL 2025.

---

### Note · Authors · 2026-01-19

I have read and agree with the venue's withdrawal policy on behalf of myself and my co-authors.